# Evaluating large language models for ADHD education: A comparative study of ChatGPT 5, DeepSeek V3, and Grok 4

Xingmin Han[1]*, Ruirui Xing[1], Mi Zhou[2]*

**1** Institute for Health and Sport, Victoria University, Melbourne, Australia, **2** Alliance for Research in Exercise, Nutrition and Activity, University of South Australia, Adelaide, Australia

* xingmin.han@live.vu.edu.au (XMH); mi.zhou@mymail.unisa.edu.au (MZ)

## Abstract

### Background

Children with attention-deficit/hyperactivity disorder (ADHD) often face barriers to participating in organized sports, particularly when physical education (PE) is delivered by outsourced coaches with limited training in disability inclusion. Meanwhile, large language models (LLMs) such as ChatGPT, DeepSeek, and Grok are increasingly used to generate educational content, yet their readability, stability, and accuracy for non-specialist educators remain unclear.

### Methods

This study systematically compared three advanced LLMs, ChatGPT 5, DeepSeek V3, and Grok 4, using identical prompts related to ADHD definitions, symptoms, and medication–exercise interactions. Thirty responses per model were collected. The primary endpoints were content accuracy, readability, and response stability. Readability was evaluated using the Flesch–Kincaid Reading Ease score (FKRE), the Flesch–Kincaid Grade Level (FKGL), and the Simple Measure of Gobbledygook (SMOG), alongside measures of lexical complexity.

### Results

All models aligned with DSM-5 in describing ADHD but differed in emphasis and stability. DeepSeek V3 produced the broadest and most variable outputs, Grok 4 showed the greatest consistency and clinical structure, and ChatGPT 5 generated concise and strengths-based explanations. However, all models exhibited high reading levels (FKGL > 12, FKRE < 40, SMOG > 12), exceeding recommended public-health readability standards for general audiences, which are typically around Grade 6–8.

**Data availability statement:** All relevant data are within the manuscript and its Supporting information files.

**Funding:** The author(s) received no specific funding for this work.

**Competing interests:** The authors have declared that no competing interests exist.

**Abbreviations:** ADHD, Attention-deficit/hyperactivity disorder; AI, Artificial Intelligence; FKRE, Flesch–Kincaid Reading Ease; FKGL, Flesch–Kincaid Grade Level; LLM, Large Language Model; PEMs, Patient Education Materials; SMOG, Simple Measure of Gobbledygook.

## Conclusion

While LLMs demonstrate strong potential for generating ADHD-related educational materials, their current readability and stability limitations restrict accessibility for non-specialist educators. Future work should focus on optimizing prompt design and language calibration to enhance usability in inclusive education contexts.

## Introduction

In recent years, increasing attention has been paid to inclusive PE, which aims to ensure that students with diverse learning and behavioral needs can participate meaningfully in physical activity. Inclusive PE emphasizes adapting instructional strategies, activity structures, and classroom management approaches to support students with neurodevelopmental conditions such as ADHD. Studies on attentional neurodiversity in PE suggest that effective inclusion often relies on flexible teaching strategies, differentiated instruction, and supportive peer environments that facilitate both participation and social interaction [1].

ADHD is one of the most prevalent neurodevelopmental disorders among children worldwide [2]. It is characterized by persistent patterns of inattention, hyperactivity, and impulsivity [3]. In recent years, research has increasingly highlighted the role of physical activity and exercise in mitigating ADHD symptoms. Moderate-intensity aerobic exercise has been shown to alleviate core symptoms of ADHD [4–7]. Children with ADHD may face difficulties participating in organized sports and group-based activities [8]. However, PE teachers themselves often report feeling underprepared to meet the needs of children with disabilities, including ADHD [9]. The situation in Australia is particularly noteworthy: a large proportion of primary school PE is outsourced to external providers, reaching as high as 78.1% in Victoria [10]. The competence of outsourced coaches, particularly their knowledge of ADHD, has been questioned [11].

Advances in artificial intelligence (AI) and LLMs are beginning to reshape education. LLMs are increasingly used to support teachers, clinicians, and parents in understanding developmental disorders [12–14]. Nevertheless, prior studies have primarily focused on educators with specialized training, such as early childhood teachers or special education practitioners [13,15]. Far less attention has been paid to how LLMs may support those with no prior knowledge, such as parents, after-school sports coaches or casual PE instructors.

For these groups, educational materials must be highly readable, concise, and practically oriented to guide the design and delivery of after-school activities. Therefore readability, accuracy, and response stability are critical when AI-generated materials are used by non-specialist educators. However, little research has systematically examined the readability and reliability of LLM-generated ADHD-related content for this population of educators.

Moreover, while multiple LLMs are currently available, each possesses distinct architectures, training approaches, and knowledge integration capacities, which may influence the clarity, comprehensiveness, and accessibility of the educational material

they produce. Different LLMs may produce responses with varying clarity, structure, and accessibility. For instance, ChatGPT 5 emphasizes human-like conversational adaptability [16], DeepSeek V3 prioritizes structured and analytical outputs [17], and Grok 4 is designed for rapid information retrieval and humor-infused explanations [18]. Identifying the most effective model for generating ADHD-related educational materials is crucial for guiding the future application of AI tools in inclusive education.

The present study aims to systematically evaluate the outputs of three leading LLMs, ChatGPT 5, DeepSeek V3, and Grok 4, to determine how effectively they can provide clear, accessible, and practical ADHD-related knowledge. This study serves as a preliminary investigation within a broader series of AI-related research, laying the groundwork for subsequent studies. Future work will involve interviews, application development, and empirical validation in the real-world contexts.

## Methods

### Study design

This study employed three LLMs: ChatGPT 5 [16], DeepSeek V3 [17], and Grok 4 [18] to generate educational materials. The educational context of outsourced PE coaching was selected as the testing domain. As ChatGPT 5 adapts and optimizes its responses based on the user's IP address [16], we report that all responses in this study were generated in Melbourne, Australia. The study was conducted from 7 August to 8 August 2025. The study evaluated the informational content and readability of the responses generated by LLMs. It did not assess the clinical effectiveness of therapies or the ability of LLMs to predict treatment outcomes. This study did not involve human participants, and all data analysed were generated by the AI models, so no ethic approval is needed for this study.

### Prompts

The prompts were designed to elicit accessible, user-friendly medical information from the AI system, following the approach described by Akkan and Seyyar [19] and Zhou et al [20]. The three guiding questions were shown in Table 1.

### Procedure

Each prompt was administered to the three LLMs under identical training and testing conditions [21]. For each model, prompts were presented sequentially to ChatGPT 5, DeepSeek V3 and Grok 4. The browser cache was cleared and a new chat session was initiated, in line with best practices for minimizing memory and context carryover [22]. All sessions were conducted on the same computer using a custom user account within a newly created virtual machine to ensure consistency of testing conditions [23]. No additional questions or distracting content were introduced beyond the pre-defined prompts. The collected responses were copied into unformatted plain text files for subsequent analysis. To capture variability across runs, ten independent attempts were generated for each question, yielding a total of 30 responses per model (10 per question). Each question was entered independently, and the three models provided responses under identical conditions, without any additional clarifying prompts or contextual information.

To ensure reproducibility and fairness across models, the exact prompt set and prompt order were kept identical for all models. The prompts were entered sequentially in the following order: Q1, Q2, and Q3. All responses were generated using the default generation parameters provided by each platform. No manual adjustments were made to

**Table 1. Questions for ADHD information.**

| Prompt |
| --- |
| Q1: *What is ADHD?* |
| Q2: *What are its symptoms?* |
| Q3: *What medications are used for ADHD, and do they conflict with physical exercise?* |

parameters such as temperature, top-p, or other sampling settings. Each prompt was submitted in a new chat session after clearing the browser cache to minimize contextual carryover. The same device and internet environment were used for all tests.

## Measures

### Quality analysis

A qualitative content analysis was conducted by summarizing and comparing the presence of predefined key content elements within each response across four dimensions: language level, information depth, structural clarity, and additional content.

Content accuracy was evaluated based on consistency with the diagnostic criteria and clinical descriptions outlined in the DSM-5. Specifically, responses were assessed for the correctness and completeness of information related to ADHD definition, symptoms, and commonly used treatments.

Two independent reviewers (XMH and MZ) evaluated all responses using this predefined evaluation framework. Each reviewer assessed the responses separately. Discrepancies between reviewers were resolved through discussion with a third researcher (RRX) until consensus was reached.

### Readability

The readability of the responses was assessed using three well-established metrics: the FKRE, the FKGL, and the SMOG [24,25]. The FKRE assigns a numerical score ranging from 0 to 100, with higher values indicating simpler and more accessible text. A score near 100 suggests that the content is very easy to understand, whereas lower scores denote increased complexity. The FKGL is an adaptation of the FKRE that estimates the minimum education level required for comprehension, expressed in U.S. school grade levels [24]. A higher FKGL score corresponds to more complex text, implying that individuals with lower levels of formal education may find the material difficult to understand [24]. The SMOG index also estimates the years of education needed for comprehension, but it focuses on the frequency of polysyllabic words (three or more syllables) within a standard 30-sentence sample [25]. Higher SMOG scores indicate greater complexity and are particularly sensitive to technical or specialized vocabulary. Number and percentage of complex words in each response were also reported. All readability metrics were calculated using the WebFX online readability test [26]. These readability indices were selected because they are widely used in health communication and patient education research to estimate linguistic accessibility of health-related texts, including studies evaluating AI-generated patient education materials (PEMs) [20,27]. Using multiple indices allows a comprehensive assessment of readability, as each metric captures different aspects of textual complexity. In general, FKRE scores above 60 indicate acceptable readability, while FKGL and SMOG scores of 6 or below are considered suitable for the general population [28].

### Data analysis

A qualitative content analysis was conducted in a descriptive manner. The results for each model were summarized concisely in tabular form, while detailed analyses of individual responses are provided in the Supporting Information (S1 File). The quantitative analysis included the calculation of readability indices (FKRE, FKGL, and SMOG), along with descriptive statistics for the number and proportion of complex words. All quantitative metrics for readability and lexical complexity were recorded in a structured database Table S1 (S2 File). Prior to statistical analysis, the distribution of quantitative variables was assessed using the Shapiro-Wilk test to examine normality. All statistical analyses and visualizations were performed in R (version 4.3.2). Descriptive statistics and line plots were used to compare results across attempts, questions, and models.

## Results

### Content analysis

Table 2 illustrates the characteristics of the three models in their responses regarding the definition of ADHD (Q1). All three defined ADHD as a neurodevelopmental disorder characterized by inattention, impulsivity, and hyperactivity, and each referred to the three DSM-5 presentations: Predominantly Inattentive, Predominantly Hyperactive-Impulsive, and Combined Type. This is consistent with DSM-5 [3]. However, clear differences were observed among the three models. DeepSeek V3 highlighted clinical evaluation procedures (psychiatric/psychological assessments, symptom checklists, and multi-informant reports) and stressed the importance of ruling out other conditions. Grok 4 explicitly listed DSM-5 diagnostic requirements, including symptom persistence for ≥6 months, onset before age 12, and occurrence across multiple settings. ChatGPT 5, by contrast, gave less attention to diagnostic criteria and instead emphasized genetic and neurotransmitter mechanisms, as well as differences in ADHD presentation between children and adults.

Moreover, both DeepSeek V3 and Grok 4 underscored a multimodal approach involving medication, behavioural therapy, and lifestyle adjustments. ChatGPT 5 concluded by emphasizing that individuals with ADHD may also demonstrate creativity and vitality. Supplementary features further distinguished the models: DeepSeek V3 included a "Myths vs. Facts" section to address societal misconceptions, Grok 4 provided epidemiological data (5–10% of children, 2–5% of adults) and noted underdiagnosis among females, and ChatGPT 5 highlighted the potential strengths of ADHD (creativity, energy, innovation) while suggesting a narrative approach to capture the lived experiences. A detailed iteration-by-iteration comparative analysis of how each model defined ADHD across the ten attempts can be found in Table S1 (see S1 File).

Table 3 illustrates the characteristics of the three models in their responses regarding ADHD symptoms (Q2). All three models described ADHD symptoms in terms of inattention and hyperactivity/impulsivity, consistent with DSM-5 diagnostic criteria. Grok 4 explicitly cited DSM-5 as its reference point, and each model elaborated on the two core symptom domains.

DeepSeek V3 included a dedicated section on "ADHD in Adults," highlighting chronic procrastination, disorganization, emotional instability, and impulsive spending as common features. Grok 4 emphasized developmental differences, noting that hyperactivity in children often manifests as running or climbing, whereas in adults it presents as "inner restlessness." It also provided examples of comorbidities and daily life impacts, such as overlap with anxiety and depression, children forgetting homework, and adults making impulsive financial decisions. ChatGPT 5 similarly stressed that adult hyperactivity is more likely to manifest as psychological unease rather than overt physical activity. The specific nuances in language level and information depth for all symptom-related responses are detailed in Table S2 (see S1 File).

Table 2. Comparative analysis of the 10 responses to Q1: What is ADHD?.

| Dimension | ChatGPT 5 | DeepSeek V3 | Grok 4 |
|---|---|---|---|
| **Language Level** | Clear, concise, academic; strictly follows DSM-5 terminology. | Clear but more expansive, blending clinical and practical language. | Accessible, straightforward; includes prevalence data that makes content relatable. |
| **Information Depth** | Focused mainly on DSM-5 definition and three subtypes. | Went beyond definition: added diagnosis, treatment, adult ADHD, Asperger's (Times 3), and hyperlinks (Times 1). | Covered treatment and prevalence rates; offered statistics for both adults and children. |
| **Structural Clarity** | Well-structured: definition, symptoms, subtypes. | More layered, combining definition with broader context; some answers lengthy but rich. | Fairly structured, but prevalence emphasis sometimes overshadowed the definition. |
| **Additional Content** | Minimal; remained close to DSM-5 only. | Provided treatment details, adult manifestations, Asperger's mention, and external resources. | Included prevalence statistics (children & adults) along with treatment, but no external resources. |

**Table 3. Comparative analysis of the 10 responses to Q2: What are the symptoms of ADHD?.**

| Dimension | ChatGPT 5 | DeepSeek V3 | Grok 4 |
|---|---|---|---|
| Language Level | Clear and clinical; also touched on emotional/executive terms accessible to wider audiences. | Detailed and descriptive, with a mix of clinical and practical expressions. | Direct, DSM-5-based language; precise and concise. |
| Information Depth | Focused on DSM-5 domains (inattention, hyperactivity/impulsivity) + added emotional and executive function challenges. | Expanded to include adult ADHD symptoms and practical advice on when to seek help. | Covered DSM-5 domains thoroughly; differentiated symptoms in children vs. adults. |
| Structural Clarity | Structured around DSM-5 core domains but broadened with extra aspects (emotion, executive function). | Rich but more layered; combined DSM-5 with real-life implications and adult cases. | Highly systematic; mapped symptoms directly to DSM-5 and split by age group. |
| Additional Content | Highlighted commonly observed but non-core issues (emotional, executive challenges). | Added practical guidance for help-seeking and adult-specific symptom detail. | Explicitly cited DSM-5; emphasized developmental differences (child vs. adult). |

Table 4 illustrates the characteristics of the three models in their responses regarding whether ADHD medications interfere with physical exercise (Q3). All three models classified medications into stimulants and non-stimulants. They differed, however, in the level of classification detail, explanations of exercise-related risks, and the extent of supplementary information. DeepSeek V3 and Grok 4 provided the most comprehensive pharmacological breakdown, whereas ChatGPT 5 presented a more concise classification. All models explained the mechanisms of action and emphasized that ADHD medications generally do not conflict with physical exercise, though certain risks should be considered. Each recommended monitoring heart rate and blood pressure, along with maintaining adequate hydration. For a comprehensive breakdown of pharmacological categories and exercise risk warnings provided by each model, refer to Table S3 (see S1 File).

Differences emerged in the scope of their recommendations. Grok 4 offered authoritative but relatively brief advice, acknowledging the benefits of exercise for ADHD management and suggesting a few activities. DeepSeek V3 provided the most detailed and pragmatic guidance, including specific exercise recommendations (e.g., jogging, swimming, yoga, strength training) and cautionary notes for high-risk activities (e.g., marathons, extreme HIIT, heat exposure). ChatGPT 5, by contrast, did not include direct exercise-related recommendations, restricting its discussion to medication effects and general safety considerations.

**Table 4. Comparative analysis of the 10 responses to Q3: What medications are used for ADHD, and do they conflict with physical exercise?.**

| Dimension | ChatGPT 5 | DeepSeek V3 | Grok 4 |
|---|---|---|---|
| Language Level | Clinical, concise; focused mainly on pharmacological terms. | Clear and explanatory; balanced technical and practical language. | Accessible and practice-oriented; user-friendly tone. |
| Information Depth | Focused mainly on medication safety, hydration, and cardiovascular monitoring, with limited direct exercise-program recommendations. | Most detailed: covered stimulants vs. non-stimulants, specific exercise types, and detailed precautions. | Provided stimulant vs. non-stimulant categories; mentioned exercise benefits and gave a few activity suggestions (though briefly). |
| Structural Clarity | Organized around medication classes and safety considerations. | Structured comprehensively: medications, exercise recommendations, safety monitoring. | Clear but less systematic; integrated medication info with brief exercise guidance. |
| Additional Content | Provided general exercise-safety advice, but offered fewer activity-specific recommendations than DeepSeek V3 and Grok 4. | Added practical advice on exercise selection and monitoring (HR, BP, hydration). | Linked medication use with exercise benefits, suggested some activities, emphasized general monitoring. |

## Readability and complexity

The model that exhibited the greatest variability was DeepSeek. Despite its average performance being in the mid-range, its fluctuations were pronounced in both readability and lexical complexity (Fig 1). The individual scores for each of the 30 iterations per model across all research questions are provided in the raw data supplement Table S1 (S2 File). Qualitative differences in the specific wording that contributed to these readability scores can be examined in the complete response logs in S4–S6 (S1 File).

In the FKRE, results across ten trials for Q1 ranged from 5.2 [95% CI: 3.8–6.7] to 27.6 [95% CI: 24.1–31.2], reflecting considerable instability, particularly in readability. In Q3, the score dropped sharply to –23.4 [95% CI: –26.9 to –19.8] before rising again to 5.1 [95% CI: 3.2–7.0]. The Number of Complex Words ranged from 115 to 229, while the Percentage of Complex Words remained between 30–38%.

ChatGPT 5 showed the second highest level of fluctuation, following DeepSeek V3. Its FKRE in Q1 ranged from 12.5 [95% CI: 11.0–14.1] to 19.8 [95% CI: 18.2–21.3]. Moreover, across the FKGL, Gunning Fog Score, and SMOG Index, scores consistently increased in complexity, culminating in Q3 with a SMOG Index of 25.4 [95% CI: 23.7–27.0]. This indicates a steadily professional tone, though at the cost of reduced accessibility. In terms of lexical complexity, ChatGPT 5 was comparatively stable, with the Number of Complex Words ranging from 70–110 and the Percentage of Complex Words holding steady at 22–28%.

In contrast, Grok 4 was the most stable of the three models. Although its scores were consistently the lowest, with FKRE in Q2 ranging from 23.6 [95% CI: 21.8–25.5] to 32.2 [95% CI: 29.1–35.4], this suggests that its professional depth was somewhat lower than the other models. Nonetheless, its stability prevented the sharp readability swings observed in DeepSeek V3. Interestingly, Grok 4 presented the highest lexical complexity in Q3, with the Number of Complex Words peaking at approximately 230.

## Discussion

### Overview of research findings

All three models aligned with DSM-5 in defining ADHD and describing core symptoms but differed in scope, depth, and style. DeepSeek V3 was the most expansive, offering broad contextual details and practical guidance. Grok 4 was the most clinically structured and stable, closely following DSM-5 but with less breadth. ChatGPT 5 was concise and accessible, highlighting strengths based perspectives and emotional dimensions while showing moderate complexity. Readability analyses further showed that DeepSeek V3 had the greatest variability, ChatGPT 5 displayed steadily increasing complexity, and Grok 4 remained the most stable and comparatively less complex.

### Comparison with previous studies

Berrezueta-Guzman et al. [13] showed that ChatGPT 5 can provide personalized guidance for children with ADHD, particularly by improving empathy, adaptability, and communication in simulated clinical environments. Similarly, Fuermaier and Niesten [29] assessed GPT-based responses in simulated ADHD scenarios, confirming their alignment with DSM-5 diagnostic standards and their ability to deliver structured definitions, symptom descriptions, and complementary recommendations. The findings of this study are consistent with these studies, emphasizing the increasing role of LLMs in inclusive education and support. Our findings further extend the existing literature by evaluating multiple LLMs under identical conditions to examine the stability of response generation. This metric is particularly important when considering large-scale deployment, especially in contexts where end-users may lack the capacity to discern factual inaccuracies. Our results indicate that Grok 4 demonstrated the highest stability, suggesting its potential utility for educational settings. However, like ChatGPT 5, it is subject to access restrictions. Frequent use within a single day requires a paid subscription, which may increase implementation costs. By contrast, DeepSeek V3 remains freely accessible.

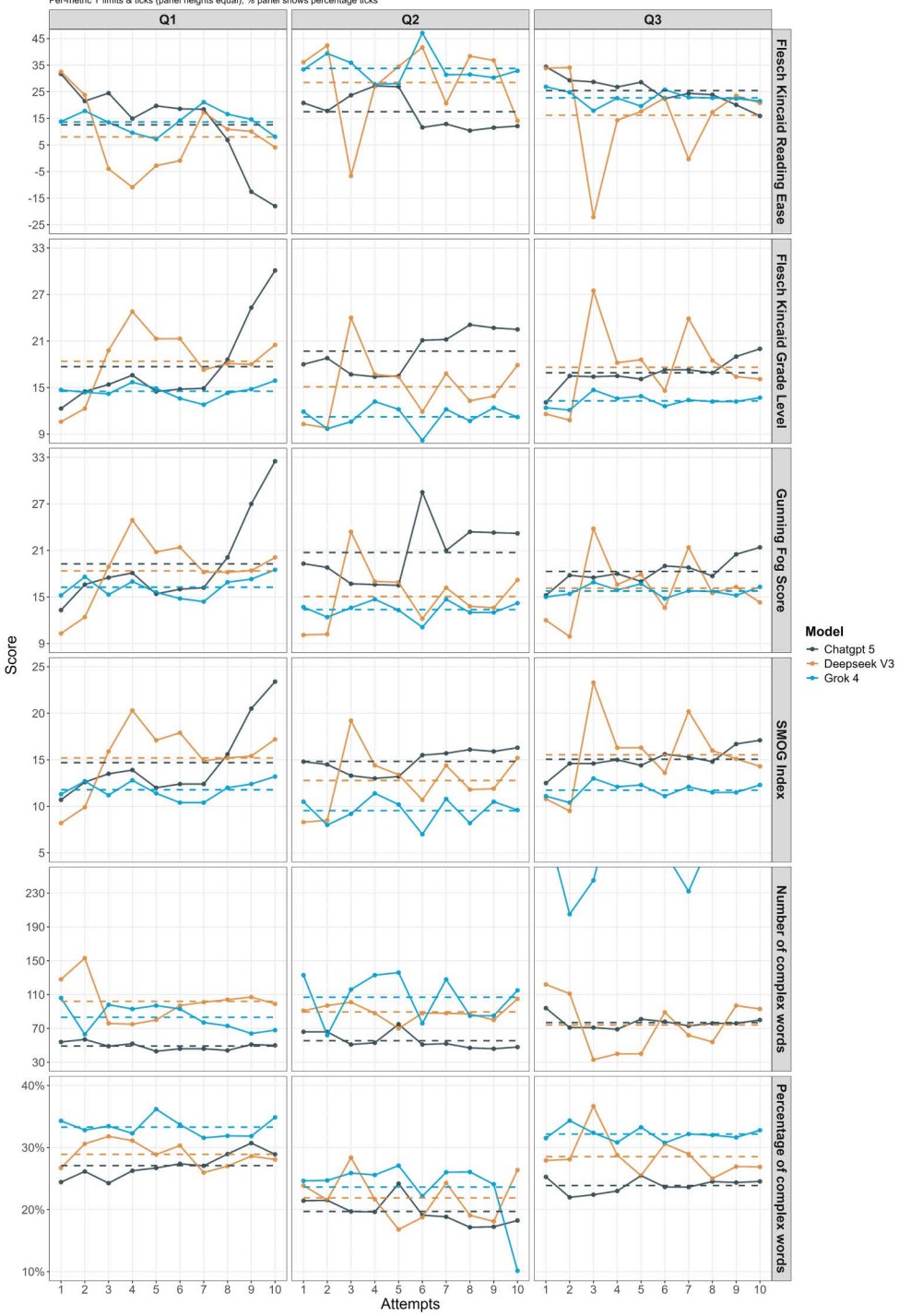

**Fig 1. Comparison of readability metrics across ChatGPT 5, DeepSeek V3, and Grok 4.** Line plots illustrate fluctuations in FKRE, FKGL, and SMOG across ten independent attempts for each question (Q1–Q3). Error bars represent 95% confidence intervals (CI).

Despite these advancements, the results also expose significant challenges. Consistent with earlier studies, we found that readability remains a key obstacle to the real-world use of LLM-generated materials [30,31]. Previous studies consistently demonstrate that AI-generated PEMs are written well above recommended public-health reading levels (Grade 6–8). For example, Beyoğlu, Kaya [32] reported that ChatGPT-generated cancer education materials required a FKGL of 15.11 ± 7.36 and a Gunning Fog Index of 18.16 ± 7.40, implying that university-level reading skills are needed for comprehension. Similarly, Akkan and Seyyar [19] found that PEMs on fragility fractures generated by ChatGPT 5 exhibited FKRE scores of 23.60–34.35 and FKGL values of 12.05–14.50, again exceeding the recommended range. When compared with these benchmarks, the outputs generated in our study demonstrate similar or even higher levels of complexity. ChatGPT 5 produced highly technical responses, with a SMOG Index of 25.4 [95% CI: 23.7–27.0], significantly higher than the 14–18 range commonly reported in prior studies. DeepSeek V3 showed the largest variability (FKRE = 5.2–27.6), shifting between clinical and conversational styles, while Grok 4 delivered the most consistent readability (FKRE = 23.6–32.2), although still above ideal levels for non-specialist users. This may constrain the practical application of these models in non-specialist educational settings. Developing responses that are accessible and easily understood may become a key focus of future research.

## Implications and future research

This study highlights both the potential and the limitations of LLMs in generating health-related content on ADHD. First, while all three models demonstrated alignment with DSM-5 in defining ADHD and describing its symptoms, their distinct emphases underscore how different design choices shape the scope and style of outputs. This suggests that model selection should be tailored to specific use cases: Grok 4 for clinically stable and structured responses, DeepSeek V3 for broad contextual guidance, and ChatGPT 5 for concise and strengths-based perspectives. Second, the findings demonstrate that response-generation stability varied between models. Grok 4's relatively high stability positions it as a promising candidate for educational and clinical support contexts where consistency is vital, though its subscription model raises concerns about cost and accessibility. Conversely, DeepSeek V3 offers open access but with significant variability, which may limit reliability. Third, readability emerged as a persistent barrier across all models, with outputs exceeding recommended public-health literacy levels. This reinforces concerns raised in prior research that LLM-generated PEMs may be inaccessible to the general public. Without improved accessibility, the broad dissemination of LLM-generated health information risks exacerbating health inequities. From a practical perspective, non-specialist educators and parents using LLMs to obtain ADHD-related information may benefit from simple prompting strategies to improve clarity and accessibility. For instance, prompts that explicitly request explanations in plain language may help reduce linguistic complexity. Users may also ask the model to provide short summaries or classroom-based examples, which can translate clinical descriptions into more practical educational guidance. Such prompt adjustments may help bridge the gap between technical health information and the needs of non-specialist audiences.

Future research should investigate strategies to optimize the readability of LLM-generated health content, ensuring alignment with recommended public health literacy standards. Comparative studies across different languages and cultural contexts are needed to evaluate whether stability and complexity patterns are consistent beyond English language outputs. Moreover, systematic experiments on fine-tuning, prompt engineering, and adaptive decoding strategies may help to balance stability with accessibility. Finally, longitudinal studies examining user comprehension, trust, and behavioural outcomes when interacting with LLM-generated materials would provide critical evidence for their safe integration into clinical and educational settings.

## Strengths and limitations

This study's primary strength lies in its comparative design, which systematically tested three of the most advanced LLMs (ChatGPT 5, DeepSeek V3, Grok 4) using identical prompts and conditions. By incorporating multiple readability

and lexical complexity indices, the analysis provided nuanced insights into both inter-model differences and intra-model variability. These findings offer practical guidance for different user groups, including teachers, parents, and healthcare providers who may have diverse needs when consulting AI-generated information about ADHD.

However, several limitations must be acknowledged. First, the study was conducted exclusively in English within a Melbourne-based testing environment, limiting generalizability to non-English-speaking populations. Second, the prompts were predefined and not iteratively adjusted, which may have constrained the models' capacity to generate adaptive responses. A further limitation is that this study did not assess whether LLM-generated responses can predict treatment outcomes or reflect the effectiveness of therapies. Future research should examine the extent to which LLM-generated information aligns with evidence-based treatment recommendations and whether it may have value in supporting treatment-related decision-making. Finally, the study focused primarily on textual readability and stability, rather than evaluating real-world comprehension or decision-making by specific user groups such as PE teachers.

Another limitation relates to potential model bias and temporal variability in large language model outputs. Because LLMs are trained on large and evolving datasets, their responses may reflect biases present in training data or system design. In addition, responses generated by these models may vary across sessions or change over time as the underlying models are updated by developers. Although this study attempted to minimize contextual carryover by using separate sessions and standardized prompts, the findings should be interpreted as a snapshot of model performance during the study period rather than a fixed representation of model capabilities. Another limitation concerns the use of traditional readability metrics to assess the accessibility of AI-generated health information. Indices such as the FKRE, FKGL, and SMOG primarily measure surface linguistic characteristics, including sentence length and word complexity. While these indicators are widely used in health communication research, they may not fully capture conceptual difficulty, contextual interpretation, or the actual comprehension of health information by readers. Therefore, readability scores should be interpreted as approximate indicators of textual complexity rather than definitive measures of user understanding.

## Conclusion

In summary, this study demonstrates that while all three LLMs aligned with DSM-5 in defining and describing ADHD, they differed markedly in scope, stability, and readability. Grok 4 showed the greatest stability, DeepSeek V3 offered the broadest contextual detail, and ChatGPT 5 emphasized accessibility and strengths-based perspectives. However, the consistently high reading levels across models highlight a major barrier to practical application, underscoring the need for future work on improving accessibility and tailoring outputs to non-specialist audiences.

## Supporting information

**S1 File. Supporting data and raw model responses.** This file contains the qualitative comparative analysis tables (Tables S1–S3 in S1 File) and the complete raw responses generated by ChatGPT 5, DeepSeek V3, and Grok 4 for all three research questions (S4-S6), providing the qualitative foundation for the study's comparative analysis.
(DOCX)

**S2 File. Minimal data set.** This file contains the structured quantitative data used for statistical analysis, including individual scores for readability indices (FKRE, FKGL, SMOG, etc.), lexical complexity metrics, and qualitative error coding for each model adherence to DSM-5 guidelines and additional content reporting across all iterations.
(DOCX)

## Acknowledgments

The authors gratefully acknowledge the developers of ChatGPT 5; DeepSeek V3; Grok 4 for providing the models used in this study, and thank all researchers whose prior work informed this evaluation.

## Author contributions

**Data curation:** Mi Zhou.

**Investigation:** Xingmin Han.

**Methodology:** Xingmin Han, Ruirui Xing, Mi Zhou.

**Resources:** Mi Zhou.

**Software:** Xingmin Han.

**Supervision:** Mi Zhou.

**Writing – original draft:** Xingmin Han, Mi Zhou.

**Writing – review & editing:** Xingmin Han, Ruirui Xing, Mi Zhou.

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
