## [Decision Letter · Decision Letter 0]

11 Feb 2026

PONE-D-25-54718Evaluating Large Language Models for ADHD Education: A Comparative Study of ChatGPT 5, DeepSeek V3, and Grok 4PLOS One

Dear Dr. Han,

Thank you for submitting your manuscript to PLOS ONE. After careful consideration, we feel that it has merit but does not fully meet PLOS ONE’s publication criteria as it currently stands. Therefore, we invite you to submit a revised version of the manuscript that addresses the points raised during the review process.

We look forward to receiving your revised manuscript.

Kind regards,

Thiago P. Fernandes, PhD

Academic Editor

PLOS One

Journal Requirements:

4. Please ensure that you refer to Figure 1 in your text as, if accepted, production will need this reference to link the reader to the figure.

Additional Editor Comments:

Please respond to all comments and highlights the changes in the ms.

Reviewer's Responses to Questions

**Comments to the Author**

1. Is the manuscript technically sound, and do the data support the conclusions?

Reviewer #1: Yes

Reviewer #2: Yes

2. Has the statistical analysis been performed appropriately and rigorously? 

Reviewer #1: Yes

Reviewer #2: Yes

3. Have the authors made all data underlying the findings in their manuscript fully available?

Reviewer #1: Yes

Reviewer #2: Yes

4. Is the manuscript presented in an intelligible fashion and written in standard English?

Reviewer #1: Yes

Reviewer #2: No

5. Review Comments to the Author

Reviewer #1: Interesting and provocative paper.

Some issues.

Abstract: primary end point should be added

Abstract: definition of high reading level should be added

Introduction for a paper like this should be shortened to 1 page

Stat methods: normal distributriuon shiuld be checkef for

methods: it is not clear who reviewed the models

Methiods: do these results of LLM predicted also response to therapies?

Reviewer #2: The study is promising, but clearer methodological rigor, stronger grounding in ADHD education literature, and deeper comparative analysis are needed.

1) Provide the exact prompt set, prompt order, temperature or generation settings, and session controls to ensure reproducibility and fairness across models.

2) Clarify how “content accuracy” was evaluated—who rated it, using which rubric, and whether inter-rater agreement was measured.

3) Expand the literature review on ADHD inclusion in physical education and coaching contexts, including established accessibility and universal design frameworks.

4) While the study provides a useful comparison of LLM-generated ADHD educational material, the related work would benefit from referencing existing research on AI-based clinical decision support and automated interpretation of physiological data. This will provide relevant insights into the challenges of accuracy, stability, and accessibility when AI systems are used to communicate health-related information to non-expert users. I suggest authors consider citing; 'Spirometer and sEMG respiratory patterns for clinical decision support system'. ( doi: 10.1109/I2MTC53148.2023.10175925 )

5) Include statistical testing (e.g., variance analysis, confidence intervals, or effect sizes) to support claims about stability and differences between models.

6) Address potential model bias and temporal variability (responses may change across sessions or updates).

7) Strengthen the practical implication section by translating findings into concrete guidance for non-specialist educators (e.g., recommended prompt templates or readability calibration strategies).

8) Justify the choice of readability metrics and discuss their known limitations for health education materials.

6. PLOS authors have the option to publish the peer review history of their article (what does this mean?). If published, this will include your full peer review and any attached files.

Reviewer #1: **Yes:** Fabrizio d'ascenzo

Reviewer #2: No

---

## [Author Response · Author response to Decision Letter 1]

26 Apr 2026

We thank the editor and reviewers for their careful reading of our manuscript and for their constructive comments. We have revised the manuscript accordingly and addressed all points raised.

A detailed, point-by-point response to all comments is provided in the Response to Reviewers document.

---

## [Decision Letter · Decision Letter 1]

18 May 2026

Evaluating large language models for ADHD education: a comparative study of ChatGPT 5, DeepSeek V3, and Grok 4

PONE-D-25-54718R1

Dear Dr. Han,

We’re pleased to inform you that your manuscript has been judged scientifically suitable for publication and will be formally accepted for publication once it meets all outstanding technical requirements.

Kind regards,

Thiago P. Fernandes, PhD

Academic Editor

PLOS One

Additional Editor Comments (optional):

Reviewers' comments:

Reviewer's Responses to Questions

**Comments to the Author**

1. If the authors have adequately addressed your comments raised in a previous round of review and you feel that this manuscript is now acceptable for publication, you may indicate that here to bypass the “Comments to the Author” section, enter your conflict of interest statement in the “Confidential to Editor” section, and submit your "Accept" recommendation.

Reviewer #1: All comments have been addressed

Reviewer #2: All comments have been addressed

2. Is the manuscript technically sound, and do the data support the conclusions?

Reviewer #1: (No Response)

Reviewer #2: Yes

3. Has the statistical analysis been performed appropriately and rigorously? 

Reviewer #1: (No Response)

Reviewer #2: Yes

4. Have the authors made all data underlying the findings in their manuscript fully available?

Reviewer #1: (No Response)

Reviewer #2: Yes

5. Is the manuscript presented in an intelligible fashion and written in standard English?

Reviewer #1: (No Response)

Reviewer #2: Yes

6. Review Comments to the Author

Reviewer #1: (No Response)

Reviewer #2: (No Response)

7. PLOS authors have the option to publish the peer review history of their article (what does this mean?). If published, this will include your full peer review and any attached files.

Reviewer #1: **Yes:** fabrizio d'ascenzo

Reviewer #2: No

---

## [Editor Report · Acceptance letter]

PONE-D-25-54718R1

PLOS One

Dear Dr. Han,

I'm pleased to inform you that your manuscript has been deemed suitable for publication in PLOS One. Congratulations! Your manuscript is now being handed over to our production team.

Kind regards,

on behalf of

Dr. Thiago P. Fernandes

Academic Editor

PLOS One